CoViD-19: an automatic, semiparametric estimation method for the population infected in Italy

http://orcid.org/0000-0002-8185-2680 Fenga Livio fenga@istat.it
ISTAT , Rome , Italy
Aly Sharif
Electronic publication date: 2021 Mar 4
Publication date: 2021
Volume: 9
Electronic Location ID: e10819
Received 2020 Apr 16; Accepted 2021 Jan 2
Copyright: © 2021 Fenga
Copyright year: 2021
Copyright holder: Fenga
License: This is an open access article distributed under the terms of the Creative Commons Attribution License, which permits unrestricted use, distribution, reproduction and adaptation in any medium and for any purpose provided that it is properly attributed. For attribution, the original author(s), title, publication source (PeerJ) and either DOI or URL of the article must be cited.
License URL: https://creativecommons.org/licenses/by/4.0/

Keywords: Autoregressive metric, Covid-19, Maximum entropy bootstrap, Model uncertainty, Number of Italian people infected

Funding: The author received no funding for this work.

==============================
To date, official data on the number of people infected with the SARS-CoV-2—responsible for the Covid-19—have been released by the Italian Government just on the basis of a non-representative sample of population which tested positive for the swab. However a reliable estimation of the number of infected, including asymptomatic people, turns out to be crucial in the preparation of operational schemes and to estimate the future number of people, who will require, to different extents, medical attentions. In order to overcome the current data shortcoming, this article proposes a bootstrap-driven, estimation procedure for the number of people infected with the SARS-CoV-2. This method is designed to be robust, automatic and suitable to generate estimations at regional level. Obtained results show that, while official data at March the 12th report 12.839 cases in Italy, people infected with the SARS-CoV-2 could be as high as 105.789.

Introduction

Covid-19 epidemic has severely hit Italy, and its spread throughout Europe is expected soon. In such a scenario, the availability of reliable information related to its spread plays a significant role in many regards. In fact, many targeted measures, such as the coordination among emergency services or the implementation of operative actions (extensive or local lock-downs or even curfew) can only be efficiently taken when reliable estimates of the epidemic spread are available at the population level.

At the moment, official data on the infection in Italy are based on non-random, non-representative samples of the population: people are tested for Covid-19 on the condition that some symptoms related to the virus are present. These data can ensure a proper estimation of the number of both deaths and hospitalizations due to the virus and are crucial for the optimization of the available resources. Nonetheless, from a statistical point of view, the number of people tested positive for Covid-19 represents a simple count which is not suitable to provide a reliable assessment of the “true”, unknown, number of infected people (thereafter “positive cases” ). In addition to the strong bias components induced by this testing strategy, there is at least another major obstacle to the construction of a valid estimator: the small sample size available. These issues are considered in the available literature: Feinstein & Esdaile (1987) point out how the statistical information in many cases can contain gross violations of epidemiological principles as well as of scientific standards for credible evidence. On the other hand, a substantial corpus of theory and methods are available to epidemiologists and/or the statisticians working on the field of epidemiology—see, for example, Kahn, Kahn & Sempos (1989) and, more recently, Clayton & Hills (2013) and Lawson (2013). Therefore, a “reasonable” trade–off between goodness of the outcomes of a statistical analysis and the available data, in some cases, is the best we can hope for. In the case of the present article, the shortness of the time series of interest is simply something that, at an early stage of an epidemic, cannot be avoided. It is well known that the shortness of the time series of interest might lead to a strong bias in the asymptotic results and therefore to the construction of biased confidence intervals. However, the results obtained in this article can be considered reliable as the approach used has been specifically designed to mitigate these negative effects. To confirm that, the estimates provided by this method have been proved to be in line with those published by official entities and have been reported on a number of nationally distributed daily newspaper published in Italy.

Based on the number of the deaths and of the observed positive cases and improving on an estimation equation proposed by Pueyo (2020), this article aims at estimating the “true” number of people infected by the Covid-19 in each of the 20 Italian regions. Presently, to the best of the author’s knowledge, Puejo’s equation does not appear in the literature nevertheless its validity in the present context will be discussed later in “Data and Contagion Indicator”. In more details, the presented procedure is designed to reduce the impact of the biasing components on the parameter estimations, by employing a resampling scheme, called Maximum Entropy Bootstrap (MEBOOT) proposed by Vinod & López-de Lacalle (2009). This bootstrap method is particularly suitable in this context: as it will be outlined in the sequel it is designed to work with a broad class of time series (including non stationary ones) and—by virtue of its inherent simplicity—is able to generate bona fide replications in the case of short time series. In fact, unlike other schemes, long time series are not required. For example, in the case of the sieve bootstrap method Andre’es, Pena & Romo (2002), a lengthy series is needed in order to estimate an high order autoregressive model from which the bootstrap replications are generated. In conjunction with MEBOOT, a distance measure—based on the theory of stochastic processes and proposed by Piccolo (1990)—has been used to find pairs of similar regions. As it will be explained later, this has been done to maintain the same methodology in those cases where one of the variable employed in the model—that is, the number of deaths—was missing.

An Overview of the Proposed Method

In small data sets it is essential to save degrees of freedom (DOF) which are inevitably lost in an amount correlated with the complexity of the statistical model entertained (see, for example, Faes et al. (2009) and Barnard & Rubin (1999)). With this in mind, the proposed method is of the type semiparametric and consists of two parts: a purely non-parametric and a parametric one. The non-parametric part refers to the maximum entropy resampling method, which will be used to generate more robust estimations. On the other hand, a parametric approach has been chosen to select certain regions on the basis of a similarity function, as it will be explained at the end of the following “Data and Contagion Indicator”. While the former does not pose problems in terms of DOF, the latter clearly does. However, the sacrifice in terms of DOF is very limited as an autoregressive model of order 1 (employed in a suitable distance function, as below illustrated) has proved sufficient for the purpose. DOF—saving strategy is also the driving force behind the choice to not consider exogenous variables such as the regions geolocation or their population—for example, in a regression-like scheme—but to implicitly assume these (and other) variables are embedded in the dynamic of the time series considered.

Data and Contagion Indicator

The paper makes use of official data, published by the Italian Authorities, related to the following two variables employed in the proposed method, that is, the number ofdeaths from Covid-19 (denoted by the symbol Mt)

currently positive cases which have been recorded as a result of the administration of the test (denoted by the symbol Ct).

The data set includes 18 daily data points collected at the regional level during the period of February 24th to March 12th 2020. The total number of Italian regions considered is 20. However, one special administrative area (Trentino Alto Adige) is divided in two subregions, that is, Trento and Bolzano. Therefore, the set containing all the Italian regions—called Ω—has cardinality |Ω| = 21 (the cardinality function is denoted by the symbol |·|). Two different subsets are built from Ω that is, Ω•—containing the regions for which at least one death, out of the group of tested people, has been recorded and Ω○ (no recorded deaths). Those two sets are now specified:{Ω∙}≡Piemonte,Lombardia,Veneto,Friuli,Liguria,Emilia,Toscana,Marche,Lazio,Abruzzo,ValleAosta,Bolzano,Campania,Puglia,Sicilia

{Ω○}≡Trento,Umbria,Molise,Basilicata,Calabria,Sardegna,

where Ω ≡ Ω• ∪ Ω○. In what follows, the two superscripts • and ○ will be always used respectively with reference to the regions {r1,r2,…r15}∈Ω∙ and {s1,s2,…s6}∈Ω○. The time span is denoted as {1,2,…,T}. In the case of the regions included in the set Ω•, following Pueyo (2020), the total number of positive is estimated as follows: (1) yj,T∙=wT∗2τδ

(2) wT=CTMT

Here, wT (Eq. 2) is the ratio between the current positive cases (C) and the number of deaths (M) whereas, in Eq. (1), τ is the average doubling time for the Covid-19 (i.e., the average span of time needed for the virus to double the cases) and δ the average time needed for an infected person to die. These two constant terms have been kept fixed as estimated according to the data so far available as reported and justified in the above mentioned Puejo’s web document. They are as follows: τ = 17.3 and δ = 6.2.

By construction, Eqs. (1) and (2) are able to properly describe the spread of the virus at the population level, as they are based on the key parameters average doubling τ and time to death (δ). To make this clear, suppose a situation where τ = δ (i.e., all the subjects, in average, die the following day after the disease has been contracted). In this case, Eq. (1) reduces to y•j,T = 2 * wT, that is we will have the total positives equal to twice the mortality rate. As for the constants chosen, they appear to be in line with the data released by the Italian public authority.

The case of the regions belonging to Ω○ is more complicated. The related estimation procedure has been carried out as detailed below (the subscript t will be omitted for the sake of simplicity):

given the series sj ∈ Ω○, a series cπ ∈ Ω• minimizer of a suitable distance function—denoted by the Greek letter π(·)—is found. In symbols: (3) cπ=argmin(c∈Ω∙)π(s,c)

the estimated number of positives at the population level—already found for cπ, say Icπ—becomes the weight for which the total cases recorded for sj, are multiplied. Therefore, the estimate of the variable of interest for this case becomes (4) yj,T○=Icπ∗CsjCrj

The distance function adopted π(·) (Eq. 3), called AR-distance, has been introduced by Piccolo (2007). Briefly, this metric can be applied if and only if the pair of series of interest are assumed to be realizations of two (possibly of different orders) Autoregressive Moving Average (ARMA) models (see, e.g., Makridakis & Hibon (1997)). Under this condition, each series can be expressed as an autoregressive model of infinite order, that is, AR(∞), whose (infinite) sequence of AR parameters is denoted by {α}j∞≡α1,α2,….

Without loss of generality, the distance between the series s and c, that is, π(st, ct) (Eq. 4), under (st, ct) ∼ ARMA(α, β), being α and β respectively the autoregressive and moving average parameters, is expressed as (5) π(s,c)=(∑j=1∞αj(s)−αj(c))1/2

Equation 5 asymptotically converges under stationary condition of the autoregressive parameters, as proved in Piccolo (2010). In other words, considering for brevity only the autoregression in αj, the roots of the polynomial Φ(z):=1−∑j=1SαjzS−j must lie outside the unit circle, that is, each root zj must satisfy |z1| > 1. For other asymptotic properties the reader is referred to Corduas & Piccolo (2008). It is well known that, with small sample sizes, the asymptotic properties of the ARMA parameters tend to deteriorate and therefore the statistical model might not perform optimally. However, in the present context their use is justified at least for two reasons: firstly the ARMA models have been here employed only for the construction of a simple distance measure used to build a similarity ranking of the Italian regions. As a simple way to pick a suitable “donor” (see the explanation below), that ARMA models tend to not perform optimally in such conditions can be considered a crucial issues. The second reason refers to the fact that, epidemics are an emergency situations and the the typical case where only a few (all the more so likely to be noisy) data points are available. Finally, in order to reach stationarity and thus correctly assess the distance functions, all the models have been estimated on properly differentiated time series.

The Resampling Method

The bootstrap scheme adopted proved to be adequate for the problem at hand. Given the pivotal role played in the proposed method, it will be briefly presented. In essence, the choice of the most appropriate resampling method is far from being an easy task, especially when the identical and independent distribution (iid) assumption (used in Efron’s initial bootstrap method) is violated. Under dependance structures embedded in the data, simple sampling with replacement has been proved—see, for example Carlstein (1986)—to yield suboptimal results. As a matter of fact, iid—based bootstrap schemes are not designed to capture, and therefore replicate, dependance structures. This is especially true under the actual conditions (small sample sizes) where the selection of the “right” resampling scheme becomes a particularly challenging task. Several ad hoc methods have been therefore proposed, many of which now freely and publicly available in the form of powerful routines working under software package such as Python® or R®. In more details, while in the classic bootstrap an ensemble Γ represents the population of reference the observed time series is drawn from, in MEB a large number of ensembles (subsets), say {γ1,…,γN} becomes the elements belonging to Γ, each of them containing a large number of replicates {x1,…,xJ}. Perhaps, the most important characteristic of the MEB algorithm is that its design guarantees the inference process to satisfy the ergodic theorem. Formally, recalling the symbol |·| to denote the cardinality function (counting function) of a given ensemble of time series {xt∈γi;i=1,…,N}, the MEB procedure generates a set of disjoint subsets ΓN≡γ1∩γ1…∩γN s.t. EΓN≈μ(xt), being μ(·) the sample mean. Furthermore, basic shape and probabilistic structure (dependency) is guaranteed to be retained ∀xt,j∗⊂γi⊂Γ.

MEB resampling scheme has significant advantages over many of the available bootstrap methods: it does not require complicated tune up procedures (unavoidable, for example, in the case of resampling methods of the type Block Bootstrap) and it is effective under non-stationarity. MEB method relies on the entropy theory and the related concept of (un)informativeness of a system. In particular, the Maximum Entropy of a given density ρ(x), is chosen so that the expectation of the Shannon Information H=E(−log⁡ρ(x)), is maximized, that is, max(ρ)H=E(−log⁡ρ(x))

Under mass and mean preserving constraints, this resampling scheme generates an ensemble of time series from a density function satisfying (4). Technically, MEB algorithm can be broken down, following Koutris, Heracleous & Spanos (2008), in 8 steps. They are:a sorting matrix of dimension T × 2, say S1, accommodates in its first column the time series of interest xt and an Index Set—that is, Iind={2,3,…,T}—in the other one;

S1 is sorted according to the numbers placed in the first column. As a result, the order statistics x(t) and the vector Iord of sorted Iind are generated and respectively placed in the first and second column;

compute “intermediate points”, averaging over successive order statistics, that is, ct=x(t)+x(t+1)2,t=1,…T−1 and define intervals It constructed on ct and rt, using ad hoc weights obtained by solving the following set of equations:

(i) g(x)=1r1exp⁡([x−c1]r1);x∈I1;r1=3x(1)4+x(2)4

(ii) g(x)=1ck−ck−1; x∈(ck;ck+1),rk=x(k−1)4+x(k)2+x(k+1)4; k=1,…,T−1;

(iii) g(x)=1rTexp([cT−1−x])rT;x∈IT;rT=xT−14+3xT4

from a uniform distribution in [0,1], generate T pseudorandom numbers and define the interval Rt = (t/T ; t + 1/T) for t=0,1,…,T−1, in which each pj falls;

create a matching between Rt and It according to the following equations: xj,t,me=cT−1−|θ|ln⁡(1−pj)ifpj∈R0xj,t,me=c1−|θ||ln(1−pj)|ifpj∈RT−1

so that a set of T values {xj,t }, as the jth resample is obtained. Here θ is the mean of the standard exponential distribution;

a new T × 2 sorting matrix S2 is defined and the T members of the set {xj,t} for the jth resample obtained in Step 5 is reordered in an increasing order of magnitude and placed in column 1. The sorted Iord values (Step 2) are placed in column 2 of S2;

matrix S2 is sorted according to the second column so that the order {1,2,…,T} is there restored. The jointly sorted elements of column 1 is denoted by {xS,j,t}, where S recalls the sorting step;

Repeat Steps 1–7 a large number of times.

The Application of the Maximum Entropy Bootstrap

In what follows, the proposed procedure is presented in a step-by-step fashion.

For each time series yt∙ and yt○ the bootstrap procedure is applied so that B = 100 “bona fide” replications are available as a result, that is, y~t,b∙;b=1,2,…B and y~t,b○;b=1,2,…B;

for both the series, the row vector related to the last observation T is extracted, that is, {v○=y~T,1○,y~T,2○…y~T,B○} and {v∙=y~T,1∙,y~T,2∙…y~T,B∙};

the expected values, that is, E(v∙) and E(v○), are then extracted along with the ≈ 95% confidence intervals (CI• and CI○ ), which are computed according to the t-percentile method. In essence, through this method, suitable quantiles of an ordered bootstrap sample of t-statistics are selected and, as a result, the critical values for the construction of an appropriate confidence interval become available. A thorough explanation of the t-percentile method goes beyond the scope of this article, therefore the interested reader is referred to the excellent article by Berkowitz & Kilian (2000).

In particular, the lower (upper) CIs will be the lower (upper) bounds of our estimator while the quantities E(v∙) E(v○) are estimated through the mean operator, that is, (6) μ○=∑j=16vj○

and (7) μ∙=∑j=16vj∙

At this point, it is worth emphasizing that the procedure not only, as just seen, requires very little in terms of input data (only the time series of the positives and the deaths are required) but also can be performed in an automatic fashion. In fact, once the data become available, one has just to properly assign the time series to the subsets Ω○ and Ω• and the code will process the new data in an automatic way. The procedure is also very fast, as the computing time needed for the generation of the bootstrap samples requires—for the sample size in question—less than 2 min. Both code and data-set employed in this article have been uploaded as Supplemental Files. However, the data can also be downloaded free of charge at the following web address: https://github.com/pcm-dpc/COVID-19/tree/master/dati-regioni (the file name is dpc-covid19-ita-regioni-20200323.csv).

Empirical Evidences

In order to give the reader the opportunity to gain a better insight on the different epidemic dynamical behaviors, in Figure 1–5 the time series of the variable C (as defined in Eq. 2) is reported for each region. Note that the sudden variations noticeable in Fig. 5 (Bolzano), Fig. 4 (Valle D’Aosta) and Fig. 3 (Molise and Campania) are due to the little number of tests administrated (i.e., the denominator of the variable wT (2)) for these cases. In emergency situations the data are usually noisy, incomplete and might show large spikes, as in the case of Fig. 5.

Figure 1 Percentage ratio deaths/new cases for the following Italian regions: Piemonte, Lombardia, Veneto, Liguria and Friuli-Venezia-Giulia.

Figure 2 Percentage ratio deaths/new cases for the following Italian regions Emilia, Toscana, Marche, Lazio and Abruzzo.

Figure 3 Percentage ratio deaths/new cases for the following Italian regions: Molise, Campania, Puglia, Basilicata and Calabria.

Figure 4 Percentage ratio deaths/new cases for the following Italian regions: Sicilia, Valle d’Aosta, Sardegna.

Figure 5 Percentage ratio deaths/new cases for the following Italian regions: Bolzano, Trento, Umbria.

That said, the main result of the article is summarized in Table 1, where three estimates of the number of positives are reported by region. The regions belonging to the set Ω○ (no deaths) are in Italics whereas all the others, belonging to the set Ω•, are in a standard format. In the columns “Mean” and “Lower (Upper) Bounds”, the bootstrap estimates computed according to Eqs. (6) and (7) and the Lower (Upper) Bounds the lower (upper) bootstrap CIs are respectively reported. The column denominated “Official Cases” accounts for the number of positives cases released by the Italian Authorities, whereas the column “Morbidity” expresses the percentage ratio between μ• (6) or μ○ (7) and the actual population of each region, as recorded by the Italian National Institute of Statistics. The latter source of data can be freely accessed at the web address http://dati.istat.it/Index.aspx? DataSetCode = DCIS_ POPRES1.

Table 1 Estimation of the number of people infected from Covid-19 by Italian regions. Lower and Upper Bounds are computed through the Bootstrap t-percentile method whereas the mean values is computed as in (6) and (7). The regions belonging to the set ° are in Italics.

	Lower bound	Mean	Upper bound	Official cases	Population	Morbidity (%)	
Abruzzo	526	600	807	78	1,311,580	0.06	
Basilicata	48	54	70	8	562,869	0.01	
Bolzano	697	730	795	103	531,178	0.15	
Calabria	182	238	493	32	1,947.131	0.03	
Campania	988	1,292	2,676	174	5,801.692	0.05	
Emilia Romagna	10,980	12,299	14,897	1,758	4,459,477	0.33	
Friuli Venezia Giulia	983	1,201	2,514	148	1,215,220	0.21	
Lazio	1,485	1,680	2,089	172	5,879,082	0.04	
Liguria	1,346	1,608	1,995	243	1,550,640	0.13	
Lombardia	37,744	45,020	49,723	6,896	10,060,574	0.49	
Marche	3,151	3,891	4,593	570	1,525,271	0.30	
Molise	119	134	167	16	305,617	0.05	
Piemonte	3,216	3,703	4,217	554	4,356,406	0.10	
Puglia	490	670	1292	98	4,029,053	0.03	
Sardegna	244	278	375	39	1,639,591	0.02	
Sicilia	776	865	1,098	111	4,999,891	0.02	
Toscana	2,352	2,755	3,965	352	3,729,641	0.11	
Trento	670	764	1,028	102	541,098	0.19	
Umbria	432	481	611	62	882,015	0.07	
Valle Aosta	139	183	356	26	125,666	0.28	
Veneto	8,382	9,343	12,028	1,297	4,905,854	0.25	
Totale Italia	74,950	87,789	105,789	12,839	60,359,546	0.18	

By examining the data for the whole Country, it is clear how the data collected by the Italian Authorities on the positive cases cannot be indicative of the situation at the population level, which appear to be greater by a factor of 8. Such a consideration, straightforward from a statistical point of view, might be worth outlining as many sources of information (e.g., newspaper, TV) mainly focus on the simple count of the positive cases so that the general public might miss the magnitude of this disease. As expected, the top three regions in terms of number of infected persons are Lombardia, Emilia Romagna and Veneto, where the estimated infected population is respectively (bootstrap mean) around 45,020, 12,299 and 9,343.

On the other hand, the risk of contagion is relatively low in some regions—mostly located in the Southern part of Italy—and in the island of Sardinia.

Regarding the regions included in the subset Ω○, the application of the Piccolo distance (π) has generated the associations reported in Table 2.

Table 2 Association found between the regions belonging to Ω○ and those in Ω• according to the minimum distance π.

Ω○	Ω•	π	
Basilicata	Veneto	0.0389	
Calabria	Campania	0.6211	
Molise	Lazio	0.4212	
Sardegna	Abruzzo	0.0157	
Trento	Abruzzo	0.00186	
Umbria	Sicilia	0.01398	

Model validation

The validation of the proposed approach is very simple and exploits the official Covid 19 mortality rate (K=DEATHINFECTED) issued by the WHO, which can be considered a well recognized and authoritative source. In essence, this constant—called K-has been used to make an estimate of the number of infected people (see Formula 8). Recalling that, in Italy, each and every person whose death was considered suspicious has been tested for Covid, it can be assumed the data related to these deaths to represent a population in itself (in other words, no inference procedures needed). The mortality rate, at the time of the writing of the paper, is K = 3.4%. By applying the simple formula (8) P=DEATHK

where DEATH refers to the number of deceased people, it is possible to have a rough estimate of the total positives (P) at a population level. However, this is not the whole story. In fact, it is well known that the virus is not capable to kill a person instantly but it takes several days to do so. Therefore, Formula 8 is now rewritten to account for this temporal lag, that is, (9) Pt=DEATHt+hK

where h is the delay time, which can be easily estimated by considering the empirical correlation function at different lags. In Fig. 6 such a structure is reported until lag h = 20. As it can be noticed, the highest correlation is at the lag h = 6.

Figure 6 Contraction of the infection—and time to death: delay structure.

Recalling that in Italy the number of Covid-19 related deaths, at the date of March 12th 2020, reached the number of 2,978, by applying 9 and using h = 6, we have: 2,9780.35=85,085.71. This number is very consistent with the estimate given in the article, which is 87.789.

Even considering higher lags, that is, h = 7,8, Eq. (9) yields respectively the following number of deaths: 97,286 and 115,200. Both these results are still within the upper confidence interval given in the article (≈105,789). Shorter lags can always be considered, even though the scientific community seems to exclude them.

Additionally, to validate the number of deaths due to Covid-19, the number of deaths occurred in the first quarter of 2020 with the average number of the deaths recorded in the first quarters of the years between 2015 and 2019 have been compared. It turns out that the total number of deaths ascribable to the Covid-19 is roughly equal to the difference between these two quantities.

Conclusions

It is widespread opinion in the scientific community that current official data on the diffusion of SARS-CoV-2, responsible of the correlated disease, COIVD-19,among population, are likely to suffer from a strong downward bias. In this scenario, the aim of this article is twofold: on one hand, it generates realistic figures on the effective number of people infected with SARS-CoV-2 at a national and regional level; on the other hand, it provides a methodology representing a viable alternative to those interested to apply inference procedures on the diffusion of epidemics.

This article proposes a methodology—illustrated in “Data and Contagion Indicator”—based on simple counts, that is, the number of deaths and the number of people tested positive to the virus for Italy, to:

provide an estimation at the national and regional level of the number of infected people and the related confidence intervals;

extend Eqs. (1) and (2) to those regions exhibiting no deaths as a consequence of the contraction of the Covid-19.

The entire procedure has been written in the programing language R® and uses official data as published by the Italian National Institute of Health. The whole code is available as a Supplemental Files in the Journal’s repository.

The results obtained show that, while official data at March 12th report, for Italy, a total of 12,839 cases, the people infected with the Covid-19 could be as high as 105,789. This result, along with the estimated average doubling time for the Covid-19 (≈6.2 days), confirms that this pandemic is to be regarded as much more dangerous than currently foreseen.

As it is well known, there are many critical questions about Covid-19 that remain unanswered. One of them is related to the mortality rate K-used in Formula 8—whose accuracy should be carefully evaluated as more and more data become available. In fact, as observed by the WHO itself, K can be biased due to the denominator of the formula being not equal to the number of infected (clinical + asymptomatic).

Supplemental Information

Supplemental Information 1 R code to input data series.

The code import the data and build 21 matrices (positives and death).

Click here for additional data file.

Supplemental Information 2 Bootstrap generation and population estimation.

R code designed to bootstrapped the time series and apply the formula (1).

Click here for additional data file.

Supplemental Information 3 Data set.

Click here for additional data file.

Supplemental Information 4 Variables.

Click here for additional data file.

The author is deeply grateful to Dr. Luigi Di Landro for the generous help in the proofreading process.

Additional Information and Declarations

Competing Interests

Author Contributions

Data Availability

The author declares that they have no competing interests.

Livio Fenga conceived and designed the experiments, performed the experiments, analyzed the data, prepared figures and/or tables, authored or reviewed drafts of the paper, and approved the final draft.

The following information was supplied regarding data availability:

Data and code are freely available at GitHub:

https://github.com/pcm-dpc/COVID-19/tree/master/dati-regioni.

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
