# Peer review of "CoViD-19: an automatic, semiparametric estimation method for the population infected in Italy"

_PeerJ, doi:10.7717/peerj.10819_

## Round 0.1 · original submission · Major Revisions

The methods are valid but in agreement with the reviewers' comments justification on use of the data, several checks to the methods, justification of results will be necessary before acceptance. Please keep in mind that not addressing these issues line by line will hinder acceptance of the manuscript for publication.

Reviewer 1 ·

Basic reporting

The manuscript is clearly written, with a good structure, figure tables and raw data shared.
The literature reference and the background/context is rather poor as far as the epidemic problem is concerned. It is acceptable, on the other hand, the literature on statistics and regression w.r.t. the problem at hand.

Experimental design

In my opinion the data set at hand is so poor that the use of data-driven models is not suitable. Few data points are available and their uncertainty is unclear. Moreover the use of the relation (1), with constants fixed as in Pueyo (2020), is crucial. How the results would change if these constants would be wrong? What is the validity of relation (1)? Pueyo (2020) is a blog, it is not a paper published on a peer reviewed journal, hence the present approach is based on a relation which is not supported by a scientific publication.

My personal feeling that any useful way to use these data should be model based, possibly using other assumptions suggested by epidemiologists. The tools used in the paper, both the bootstrap strategy and the algorithm to make associations based on similarity, are correct and sound; however I doubt that the results on this data set can be trustable, because few points are available.

Validity of the findings

I think that this paper fails to provide useful results for the estimation of the number of infected persons, because the data set is poor and because use of the equation (1) is problematic. I would not recommend any data driven method on this data set.

Reviewer 2 ·

Basic reporting

Please see General comments for the author

Experimental design

Please see General comments for the author

Validity of the findings

Please see General comments for the author

Additional comments

The topic of interest in this article is timely and extremely important. In this sense, the article has the potential to answer one of the most important questions on the infection rate in the population. Please see my comments below.

Major:
1. In the abstract, it's mentioned that "The result obtained show that, while official data at March the 12th report 12.839 cases in Italy, the number of people infected with the CoViD–19 could be as high as 105.789." Please make the distinction between the number of reported infected people and the prevalence of the disease in the population. Here 12.839 is just the reported number and it's not claimed to be the estimate of the number of infected people in the population. So, please rephrase this sentence.

2. Please review the English language of the manuscript. There are some misuses of certain words.

3. Line 211-213, obviously the data published by the governments do not constitute statistical estimates of different rates in the population. They are just descriptive numbers. Thus, it is not fair to claim that "how the data collected by the Italian Authorities on the positive cases severely underestimate the current situation by a factor of about 8." This needs to be rephrased.

4. Line 28, "incorrect" is unsuitable to use with confidence interval since there is no correct confidence interval. Please consider that a confidence interval is an interval estimator.

5. The description of the proposed method in Section 2 is somewhat vague. Please rewrite this section to give a clear description of your method.

6. The validity of the ARMA assumption on the distance needs to be demonstrated. It is a quite strong assumption to expect to hold with a short time series data.

7. There is no reference given to Figure 1. Line 193, would be Figure 1-5 instead of Figure 2-5.

8. Please use a meaningful visualisation for the results presented in Table 2 and remove Table 2 from the manuscript.

9. Please include the file COVID_13_3_2020A.txt in your submission. Results are not reproducible without this file.


Minor:
1. You can consider using { } for your set definitions in Section 3.

2. you do not need to indent on line 68.

Reviewer 3 ·

Basic reporting

The overall paper is very well written. Quality of presentation, figures and tables, references cited are all satisfactory.

Minor comments:
line 24: mayor -> major

Why are the summations of Eqn 6,7 going upto 6?

line 196-197: denominator of C_T or w_T?

Experimental design

The fundamental components of the model is based on (Pueyo 2020). However this work itself is not peer-reviewed and not credible to this reviewer.

For example, in Eqn 1, proper estimates of \delta and \tau are challenging to get; why only consider average values when these terms may have large variance? The validity of Eqn 1 needs to be established properly as it forms the basis for the model proposed here.

Validity of the findings

The model depends on the currently recorded infected and death counts and does not consider the population density into account. This is somewhat captured in the average doubling time which in itself is not a good metric.

Since the model lacks credibility, all the corresponding conclusions are speculative at best. A proper validation of the model is required.

I would suggest the authors to execute a simple SIR or SEIR model using (i) the tested infected cases and (ii) the model estimates. It is expected that as time passes the model predictions come closer to the tested positives, considering the testing rates are increasing over time.

Additional comments

This paper tries to investigate a very important problem of estimated the actual numbers of infected people considering the tested positives in a region. However, the model needs rigorous validation exercises before it can be published. At its current state, the model is very speculative and makes some simplifying assumptions.

·

Basic reporting

This is an interesting paper with specific and important objectives.

Minor Comments

1. Clear, unambiguous, professional English language used throughout. However, there are many stylistic issues, as noted below. The numbers in the last two lines of the abstract seem like decimals. Please change 12.839 to 12,839 and 105.789 to 105,789.

2. Please choose more specific keywords. The current keywords such as “number of Italian people infected” or “model uncertainty” are vague.

3. Line 9: do not use the word “diffusion” as it refers to randomness of disease spread. We don’t have any evidence supporting the randomness of COVID-19 spread.
4. Lines 17,18: It is claimed that “people are tested for CoViD-19 on the 18 condition that some symptoms related to the virus are present.” Given the rapid changes in the policy and procedures related to COVID-19, can the author confirm that this is still the case.

5. Line 69: change the word being with “where”
6. Line 40: section 2 is not really the “proposed method” but an overview of the proposed method. Please

7. Lines 75 and 76: Please include index “T” for C and M

8. Line 84: the subscript should be capital T not small t

9. Please include the related references for the first two paragraphs of the introduction. Your statements must be evidenced by reliable scientific publications and data sources.


10. Equation 5, please change the notation of square root to the exponent of 1/2


11. Lines 209 and 190: please include the links as references.

12. Page 11, Table 1 should be centered.

Experimental design

N/A

Validity of the findings

Major comments

1. The resampling method mentioned in section 4 are very standard and the author prefers, they can mentioned them in an appendix. Similarly, the max entropy method can be mentioned in an appendix

2. Section 3: please include a map indicating the regions containing at least one death, and regions without any death.

3. Is equation (5) always convergent? If yes, please mention it. If no, then state the condition under which the series is convergent.

4. Lines 98, 99: Note that ARMA models work on the assumption of stationarity (i.e. they must have a constant variance and mean). You need to check if this assumption holds with your data. Specifically, check for stationarity before fitting an ARMA model---plot the data, inspect the sample ACF and PACF, pretest for unit root (ADF or other tests), difference if necessary, etc.

5. The last row of Table 2 does not seem correct. Are the numbers on the third and fourth columns decimals or commas? Similarly, the population column has decimals instead of commas. The last column should be all decimals. Please correct them all.

6. Lines 192-197: Figures 2-5 deserve a better explanation and justification. For instance, what is the justification for the huge spike in Figure 5?


7. The distance function in equation 3 is not clear to me. Are we measure the distance from the center of a region without any death to the center of the region with death? Or we a measure the center of a region without any death to the closest death that occurred. If it is the first case, then you need to indicate it below equation 3. If it is the second case, then the data description in section 3 did not really mention the details of the data.

Additional comments

In conclusion: There are a few weaknesses in the statistical analysis (as I have noted above) which should be improved upon before Acceptance.

---

## Round 0.2 · Major Revisions

Most of the reviewer comments have been addressed, however, several issues continue to threaten the success of this manuscript, specifically:

1. There is a fundamental lack of evidence that the model proposed functions in an unbiased way. The difference between observed and predicted estimates of number of positives, which by the way is 7 to 8 times in magnitude, cannot be simply attributed to biased testing. Before we even go into statistical reasons, just intuitively, public health authorities, are more likely to target test the localities with high cases, so one can argue the opposite, that the published numbers are over estimates. The latter for all reasons may or may not be true, but so is the speculation that the observed is an under estimate of the actual number of cases. Rather, a true statistical simulation with generated data of known prevalence estimates (you will need several to produce a sensitivity analysis, start with low prevalence, moderate and high) with varying degrees of bias (say 100% of actual cases, 90%, 80%, … till 10%). Now run your model and show the delta explaining its magnitude by level of bias (to make your case that the model proposed can accurately estimate prevalence, and report the accuracy level). Also the sensitivity analysis with prevalence should be able to show case what prevalence estimates your model fails at if any. The result is a 3-D plot showing what prevalence and what bias levels (underreporting or undertesting) does your model prove to be useful, and where it fails.
This isn’t obviously the only way to show case our model’s accuracy and performance, but it is a sound statistical simulation that will answer the questions raised by the reviewers and myself. Other methods I deem acceptable are use of mathematical models, a simple SIR model will produce the transmission in silico and allow you test your model likewise, however, my concern is you don’t have the parameters needed for the mathematical model (we all don’t yet, for example, what is the contact rate for COVID-19, and where, under what circumstances and when was it measured, we don’t know).

2. The equations 1 and 2 you mention they are following Puoye 2020, but its not clear if you derived these formulas yourself or they came from the medium.com published web article? (Who derived these you or the medium.com author, and where do you see them? In the weblink you provided or the google sheet document online? Also I agree with reviewer 3 that the average estimates for doubling time and time to death. This should be incorporated in your model, or at least test their importance using sensitivity analysis. Also replace “killing time” with time to death OR inverse of mortality rate.
By the way, referencing the medium.com article is not acceptable, you may refer to it in the manuscript text parenthetically though.
3. Measures of uncertainty (standard errors) are lacking through out the manuscript.

4. Several reviewer comments have not been addressed, specifically Reviewer #4 comments 3 to 7 from the previous review cycle. For each of the comments I have copy pasted below for your convenience, I didnt think your response was satisfactory and for most certainly didn’t make the case in the manuscript. Please readdress the comments below:

---------------START of reviewer comments from previous review to be addressed -------------
3. Is equation (5) always convergent? If yes, please mention it. If no, then state the condition under which the series is convergent.

4. Lines 98, 99: Note that ARMA models work on the assumption of stationarity (i.e. they must have a constant variance and mean). You need to check if this assumption holds with your data. Specifically, check for stationarity before fitting an ARMA model---plot the data, inspect the sample ACF and PACF, pretest for unit root (ADF or other tests), difference if necessary, etc.

5. The last row of Table 2 does not seem correct. Are the numbers on the third and fourth columns decimals or commas? Similarly, the population column has decimals instead of commas. The last column should be all decimals. Please correct them all.

6. Lines 192-197: Figures 2-5 deserve a better explanation and justification. For instance, what is the justification for the huge spike in Figure 5?


7. The distance function in equation 3 is not clear to me. Are we measure the distance from the center of a region without any death to the center of the region with death? Or we a measure the center of a region without any death to the closest death that occurred. If it is the first case, then you need to indicate it below equation 3. If it is the second case, then the data description in section 3 did not really mention the details of the data.
---------------END of reviewer comments from previous review to be addressed -------------

5. Several reviewers commented on grammatical errors. I went through the manuscript and identified the following:
a. Line 57: “a lengthy series in needed..” edit to is needed, if that is what you meant
b. Line 66: “a amount” should be “an amount”
c. Line 110: replace killing time with time to death
d. Line 174: “scheme has not negligible advantages” this needs to be reworded for clarity, are you saying that that scheme has no negligible advantages, can you replace with simply the scheme has important or significant advantages?

My biggest concern here is the lack of model validation (see my first comment), I don’t see how can your findings be defended without such a simulation or other validation method that is par to it, specially given that COVID isn’t not a disease that we have studied in most continents or we have data on for decades. This model validation and the comments and corrections I identified above are required for this manuscript to move beyond R1.

Reviewer 2 ·

Basic reporting

The is no problem with reporting.

Experimental design

The research question is clear.

Validity of the findings

No problem identified on the findings.

Reviewer 3 ·

Basic reporting

Presentation, language and references are fine.

Experimental design

Methods appear to be fine. However they are still speculative (See below).

Figs 1-5 are not that meaningful at all. They do not help with model validation in any way and simply show some numerical trends. These figures can very well be removed.

Validity of the findings

In the first round I asked for the following:

I would suggest the authors to execute a simple SIR or SEIR model using (i) the tested infected cases and (ii) the model estimates. It is expected that as time passes the model predictions come closer to the tested positives, considering the testing rates are increasing over time.

This comment was not addressed. The true infected as inferred by the model when incorporated in a SEIR/SIR model might be able to show the peak infections (timing and amplitude) which can serve as a model validation. It is expected that the reported infected counts will not be able to match the infection peak. This is specially meaningful in Italy where the curve has flattened already. I would recommend the authors to show a simple SIR model projection considering their estimated infected counts that should correlate with the observed peak infections in Italy.

---

## Round 0.3 · Minor Revisions

Dear Dr. Fenga,
There are a few edits I made to the text and a couple of comments where I suggest rewording, all are in the pdf I am including in my decision for minor revisions. I am also suggesting a final paragraph in the discussion that should offer further protection to your research since even the WHO reported mortality rate is based on a denominator of the population tested can be an under-estimate. Don't get me wrong, your proposed method comes even closer but one can still argue that a dataset on a population completely tested, as in a census or random survey sample is needed. This dataset probably doesn't exist, yet, but is likely to soon, as COVID-19 testing becomes streamlined. Hence, adding verbiage that indicates that while your method can be more accurate than the current stats reported, its validation using a complete population survey or complete population testing where possible can offer a complete validation of the methodology described here.

I hope you can make the editorial changes I suggested so that I can recommend full acceptance soon after.

Best wishes,
Sharif

Reviewer 1 ·

Basic reporting

Still, I find the statistical approach, and the handling of uncertainties, quite obscure for the general readership and I have doubts that precise inferences can be done out of these data. I see that the author has made some improvements in the new version but in my opinion the manuscript is not publishable yet.

Experimental design

no comment

Validity of the findings

no comment

Additional comments

Although the authors improved the paper, the revised version still fails to convince that the data analysis performed here, on these very limited data, would be interesting from the epidemiology's point of view.

Reviewer 2 ·

Basic reporting

All good with reporting.

Experimental design

Suitable

Validity of the findings

Validated appropriately.

Reviewer 3 ·

Basic reporting

The presentation of the paper is fine.

Experimental design

The methods are rigorous.

Validity of the findings

The findings appear valid. The paper now includes a validation section which can provide some confidence to the results reported here.

Additional comments

My comments have been satisfactorily addressed.

---

## Round 0.4 · accepted · Accept

Thank you for addressing the reviewers and my comments, congratulations I have accepted your manuscript.